# Advances in Understanding of Metabolism of B-Cell Lymphoma: Implications for Therapy

**DOI:** 10.3390/cancers14225552

**Published:** 2022-11-11

**Authors:** Katarina Kluckova, Annalisa D’Avola, John Charles Riches

**Affiliations:** Centre for Haemato-Oncology, Barts Cancer Institute—A Cancer Research UK Centre of Excellence, Queen Mary University of London, Charterhouse Square, London EC1M 6BQ, UK

**Keywords:** germinal center biology, B-cell metabolism, lymphoma metabolism, phosphatidylinositide-3-kinase, mammalian target of rapamycin, glutaminase, oxidative phosphorylation, monocarboxylate transporters

## Abstract

**Simple Summary:**

The role of metabolism in normal and malignant B-cell biology is an area of rapid evolution and interest. While previous research has focused on glycolysis, glutaminolysis and oxidative phosphorylation, recent advances have demonstrated the importance of additional metabolic pathways and how these interact with oncogenic drivers and cellular signaling. This article reviews the current understanding of normal germinal center metabolism and how this relates to the metabolism of germinal center-derived lymphomas. There is increasing interest in the potential of targeting metabolic pathways for anti-cancer therapy, with some new agents already entering clinical trials. This article will also review novel enzymatic targets and pathways, and how existing agents modulate metabolism.

**Abstract:**

There have been significant recent advances in the understanding of the role of metabolism in normal and malignant B-cell biology. Previous research has focused on the role of MYC and mammalian target of rapamycin (mTOR) and how these interact with B-cell receptor signaling and hypoxia to regulate glycolysis, glutaminolysis, oxidative phosphorylation (OXPHOS) and related metabolic pathways in germinal centers. Many of the commonest forms of lymphoma arise from germinal center B-cells, reflecting the physiological attenuation of normal DNA damage checkpoints to facilitate somatic hypermutation of the immunoglobulin genes. As a result, these lymphomas can inherit the metabolic state of their cell-of-origin. There is increasing interest in the potential of targeting metabolic pathways for anti-cancer therapy. Some metabolic inhibitors such as methotrexate have been used to treat lymphoma for decades, with several new agents being recently licensed such as inhibitors of phosphoinositide-3-kinase. Several other inhibitors are in development including those blocking mTOR, glutaminase, OXPHOS and monocarboxylate transporters. In addition, recent work has highlighted the importance of the interaction between diet and cancer, with particular focus on dietary modifications that restrict carbohydrates and specific amino acids. This article will review the current state of this field and discuss future developments.

## 1. Introduction

During development and differentiation, B-cells undergo significant variations in metabolic activity. In the periphery, naive B-cells are metabolically quiescent, which is thought to contribute to their long-term viability [1]. During B-cell activation, this relatively inactive metabolic state is reversed to meet the demands of growth, proliferation and antibody production [2,3]. Activated B-cells (ABCs) can either differentiate into extrafollicular plasmablasts essential for early protective immune responses or enter a germinal center (GC). The purpose of the GC reaction is to facilitate the process of B-cell receptor (BCR) affinity maturation, by rounds of proliferation and somatic hypermutation facilitated by attenuation of normal DNA damage checkpoints. This means that GC B-cells are particularly prone to transformation and most B-cell lymphomas originate at this stage, such as Burkitt Lymphoma (BL) and subtypes of diffuse large B-cell lymphoma (DLBCL) [4,5]. Alterations in metabolism are a hallmark of cancer [6], with many tumors inheriting the metabolic program of their tissue of origin [7]. Therefore, to understand the metabolism of B-cell lymphomas it is of critical importance to understand the metabolic needs of normal B-cells at different stages of their development.

This review will summarize the current understanding of normal GC metabolism, how this is reflected in observations regarding the metabolism of GC-derived B-cell lymphomas and how this is shaped by the microenvironment. It will then discuss how advances in lymphoma metabolism could be translated into novel therapies for B-cell lymphomas in the clinic.

## 2. Summary of Normal GC Metabolism

Resting normal naïve B-cells show little metabolic activity and do not proliferate [2]. On encountering their cognate antigen presented on the surface of dendritic cells within lymph node follicles, they either differentiate directly to extrafollicular plasma cells (PCs) and memory cells or form a germinal center [8,9]. B-cells recognize this antigen via their specific surface immunoglobulin receptor with this interaction triggering BCR-signaling and activation of downstream pathways including phosphoinositide 3-kinase (PI3K)/protein kinase B (Akt) activation. The proliferation and differentiation of GC B-cells also requires a second signal that is typically provided by antigen-specific T helper (Th) cells, but can be provided in a T-cell independent process via pattern-recognition receptors such as Toll-like receptors [9,10]. Initiation and formation of GCs is regulated by expression of several factors, among them MYC, B-cell lymphoma 6 (BCL6) and E2A, which are often dysregulated in GC-derived lymphomas [4,11]. Each GC consists of light (LZ) and dark zones (DZ) defined by histological appearance [12]. The darker appearance of DZs is due to densely packed proliferating B-cells which are undergoing somatic hypermutation (SHM) of their BCR. From the DZ, GC B-cells then move to the LZ populated by follicular T helper (Tfh) cells and dendritic cells. The LZ is the place of affinity testing of the newly mutated BCR and from here, the positively selected cells can either return to the DZ for another round of proliferation and SHM or directly differentiate to long-lived PCs or memory B-cells [11]. The GC exit towards differentiation is further controlled by several mechanisms, including epigenetic regulations such as downregulation of trimethylation of the lysine 27 of histone 3 (H3K27me3) by the histone methyltransferase enhancer of zeste homolog 2 (EZH2) [11]. Activation of EZH2 prevents the terminal differentiation of GC B-cells and is another known driver of lymphomagenesis [4,13].

The role that metabolism plays in GC biology is complex, with some observations that appear conflicting at first sight. BCR stimulation primes B-cells for both glycolysis and OXPHOS via PI3K/Akt and PKC beta signaling [10,14,15,16,17,18]. The earliest events after BCR engagement such as B-cell spreading and antigen processing appear to rely solely on OXPHOS, which may reflect the metabolic skewing of naïve B-cells and allow time for glycolytic reprogramming to be instigated [2]. PI3K/Akt signaling regulates metabolism on multiple levels and via various downstream transcription factors, including MYC [19]. Increased MYC expression is observed within LZ GC B-cells that are “licensed” for further rounds of proliferation within the DZ and is instrumental for the cycling between the two zones [20,21]. A further factor is that LZ B-cells also activate mTORC1 enabling adoption of an anabolic phenotype to promote glucose uptake, cell growth and ribosomal biogenesis in anticipation of proliferation [22]. In spite of being highly active, GCs are characterized by nutrient deprivation and hypoxia, particularly within the LZ [23]. Therefore, the induction of MYC-dependent metabolic changes may be counterbalanced by the action of glycogen synthase kinase 3 (GSK3), which inhibits glycolysis and mTOR signaling in GC B-cells to limit metabolic exhaustion within this demanding microenvironment [8,24]. The hypoxic environment and stabilization of hypoxia-inducible factor 1α (HIF1α) observed in LZ GC B-cells is another factor that can influence the metabolism of GC B-cells [23,24,25]. HIF transcription factors mediate the response to hypoxia to increase glucose uptake and utilization. Indeed, HIF1α-mediated glycolysis has been observed in murine and human GC B-cell subsets, with upregulation of HIF1α in the GC affecting the function of other metabolic regulators such as mTOR and MYC, both of which were implicated in cross-talk with HIF factors [23,26,27,28].

The importance of OXPHOS has been highlighted by a recent study which demonstrated that GC B-cells rely on oxidation of long-chain fatty acids [29]. This is supported by observations demonstrating that LZ and DZ GC B-cells manifest distinctive expression dynamics of OXPHOS and glycolysis, with DZ cell clusters having the highest expression of OXPHOS pathway genes, with OXPHOS implicated in positive selection and affinity maturation [30]. Notably, this report observed that in vivo glucose uptake was higher in bulk GC B-cells in comparison to resting B-cells, with this increase being more apparent in LZ cells in contrast to proliferating DZ GC B-cells where OXPHOS is more important [30].

Overall, the concept of GC B-cells having a single metabolic program is likely to be a gross oversimplification. Instead, it is likely that B-cells modulate their metabolism depending on where they are within the GC reaction, along with the relative affinity of their BCRs. Fatty acids alone, for example, cannot fully support the tricarboxylic acid (TCA) cycle without the contribution of glucose- or glutamine-derived carbons needed for net-oxaloacetate synthesis [31]. Furthermore, phosphoglycerate dehydrogenase (PHGDH), which is necessary for de novo serine synthesis, is required for GC formation [32]. As this enzyme facilitates the conversion of a glycolytic intermediate into serine, this would suggest that glycolysis is important in GC B-cells. Also, the role of MYC is likely to be important with the induction of MYC observed after positive selection in the LZ instigating a MYC-driven metabolic program which may persist into the DZ even after MYC expression is lost [33]. Changes in metabolism have also been implicated in B-cell fate with the decision to differentiate into memory or plasma cells being partially mediated by metabolic pathways. Indeed, a subset of B-cells in the GC, characterized by decreased mTORC1 and MYC-signaling, have a tendency to differentiate into memory B-cells [34,35]. In contrast, while PCs require a high level of metabolic activity to fuel extensive immunoglobulin production, they stop proliferating when terminally differentiated, having very different metabolic demands when compared with a GC B-cell. Instead, PC metabolism is characterized by upregulation of OXPHOS fueled by glutamine, with PC glycolysis supporting antibody glycosylation via the hexosamine pathway [2].

## 3. The Metabolism of Lymphomas

### 3.1. Overview

A clinically important feature of the main types of high-grade lymphoma is that they are characterized by increased glucose uptake. This is the basis for widely used clinical tests such as ^18^F-fluorodeoxyglucose-positron emission tomography (FDG-PET). Patients are injected with ^18^F-fluorodeoxyglucose, which is taken up by tumor cells and phosphorylated by hexokinase but not metabolized further, leading to its accumulation within the lymphoma cells underpinning its clinical utility. The increase in glucose uptake is paralleled by increased flux through glycolysis and increased lactate production. The significance of this is reflected in clinical observations of patients with advanced stage high-grade lymphoma having a resting metabolic acidosis and hyperlactatemia [36,37]. The enzyme which converts pyruvate to lactate, lactate dehydrogenase (LDH) is widely used in many prognostic indices for risk stratification of patients with lymphomas [38]. LDH is known to be reflective of cell turnover, but its role in maintaining glycolytic flux to regenerate nicotinamide adenine dinucleotide (NAD^+^) for anabolic biosynthesis may well be important. In addition, in vivo observations in humans using labeled lactate have provided evidence that it can be used as a fuel by cancer cells, which may also be factor in lymphoma [39].

The transcription factor MYC is well established as a master regulator of metabolism and proliferation, where it upregulates many genes involved in glycolysis, glutaminolysis and mitochondrial metabolism (Figure 1) [40]. *MYC* translocations are a hallmark of BL and MYC is expressed in almost half of DLBCL cases [41,42]. Furthermore, *MYC* translocations are a poor prognostic factor in DLBCL in association with *BCL2* and/or *BCL6* translocations, and *MYC* re-arrangements are also a driver of high-grade transformation in follicular lymphoma (FL) and chronic lymphocytic leukemia (CLL) [43]. An interesting model for the study of early MYC-linked B-cell transformation events was reported in EBV-infected primary B-cells, which observed increases in glycolysis, serine import and de novo serine synthesis, and mitochondrial one-carbon (1C) metabolism. The importance of this goes beyond EBV-driven lymphomas as MYC overexpression was able to induce similar changes in 1C metabolism even when the EBV-encoded viral factor *EBNA2* was inactivated [44].

### 3.2. Burkitt Lymphoma

BL is the prototypical example of a MYC-driven cancer. Translocation of the *MYC* oncogene placing it under the regulation of immunoglobulin loci, is present in all cases [4,45]. BL shares a transcriptional signature with DZ GC B-cells and is thought to arise at this stage of the GC reaction [4,46]. DZ GC B-cells are among the most proliferative cells in the human body, dividing every 6 h. Consistent with this, BL is an extremely proliferative cancer with more than 95% Ki67 positivity [43]. However, paradoxically, MYC is absent in normal DZ GC B-cells where its expression is thought to be suppressed by BCL6 [20,47]. One suggested explanation for this is that *MYC*-induced expression of the transcription factor AP4 is able to maintain the expression of MYC target genes in the DZ even after MYC itself is downregulated [33,48].

Unsurprisingly, MYC overexpression in BL cells results in many metabolic features known to be driven by MYC including increased glucose consumption and lactate production, but also increased contribution of glucose-derived carbon to the TCA cycle [49]. MYC also has been shown to cross-talk with HIF1α in this lymphoma. Although HIF1α has been shown to counteract the pro-proliferative effect of MYC in some non-B-cell cancers, these two factors co-operate in BL to promote glycolysis by induction of HK2 and pyruvate dehydrogenase kinase 1 [27,50]. In addition, there is also increased glutaminolysis, with tracing experiments demonstrating a significant contribution of glutamine-derived carbon into TCA cycle when MYC is overexpressed. This use of glutamine appears to help MYC-driven BL cells maintain their viability and proliferation rates even when extracellular glucose is scarce [49]. 70% of BL have also dysregulated activity of the transcription factor E2A/TCF3 due to mutations in *TCF3* or *ID3*, which results in tonic BCR signaling via PI3K [4]. Interestingly, the combination of constitutive MYC expression with PI3K activity in GC B-cells results in development of lymphoma in transgenic murine models, which is highly reminiscent of human BL [51]. From a metabolic viewpoint, tumors driven by tonic BCR signaling have recently been shown to be susceptible to inhibition of mitochondrial 1C metabolism [52]. Furthermore, our own work demonstrated the importance of serine and 1C metabolism in lymphoma development in the *Eµ-Myc* lymphoma mouse model, highlighting the therapeutic potential of inhibiting de novo serine synthesis [32].

### 3.3. Diffuse Large B-Cell Lymphoma (DLBCL)

DLBCL is the commonest form of high-grade lymphoma. DLBCL cases can be classified by their cell-of-origin as GCB-DLBCL or ABC-DLBCL [53]. GCB-DLBCLs have a transcriptomic profile resembling that of LZ GC B-cells. Intriguingly, ABC-DLBCLs have a transcriptomic profile comparable to that of B-cells activated in vitro with anti-BCR and CD40 ligation. Despite these stimuli originally being designed to mimic the conditions within the normal GC, they actually give rise to a transcriptional profile distinct from most GC B-cells, which have been more recently found to resemble a group of B-cells in the process of exiting the GC (post-germinal center plasmablasts) [4,46]. ABC DLBCLs require BCR-signaling for their survival, which is known to be chronically active in this subtype and therefore amenable to targeting with BCR inhibitors [54,55]. BCR signaling is known to upregulate glycolysis in normal B-cells, so it is no surprise to find that DLBCLs are invariably glycolytic and FDG-avid [18].

A further classification of DLBCL, independent of the cell-of-origin subtypes, relates to their metabolic signature: OXPHOS-DLBCL with upregulated mitochondrial metabolism; BCR-DLBCL with increased expression of cell-cycle regulatory genes; with a third DLBCL subset characterized by markers of an ongoing host inflammatory/immune response—HR-DLBCL [56]. The OXPHOS-DLBCL cell lines were found to be preferentially fueled by fatty acids while the BCR-DLBCL cells relied on aerobic glycolysis despite comparable rates of glucose uptake between the two subtypes [57]. Interestingly, inhibition of BCR signaling led to increased palmitate-linked respiration suggestive of a metabolic switch between glycolysis and OXPHOS related to the chronic BCR signaling in this subtype [54,57]. OXPHOS-DLBCL cells were found to be more sensitive to inhibition of mitochondrial translation pathways and downregulation of glutathione (GSH) synthesis [57,58]. On the contrary, the increased GSH levels in the OXPHOS subtype were linked to their lower sensitivity to histone deacetylase inhibitors [59]. A further study demonstrated that cases of OXPHOS-DLBCL also tended to have low expression of the glycolytic enzyme glyceraldehyde-3-phosphate dehydrogenase (GAPDH). GAPDH^low^ lymphomas also required mTORC1 signaling and glutaminolysis, with combined inhibition of all three pathways inhibiting lymphoma growth in murine models and inducing clinical responses in human DLBCL patients [60]. Interestingly, increased expression of GAPDH mRNA in biopsies from DLBCL patients correlated with higher levels of HIF1α and NFκB, and this axis was shown to promote lymphoma vascularization and aggressiveness [61].

Expression of MYC is observed in almost half of DLBCL cases including those without *MYC*-translocations and is associated with poor prognosis [42,43]. The prognostic impact of *MYC* translocations depends on presence of another hit, usually the anti-apoptotic gene BCL2, less commonly BCL6. These are known as double-hit lymphomas (DHL; or triple-hit lymphomas (THL) if all three genes are rearranged) and have unfavorable progression-free survival and overall survival [42]. In contrast high expression of MYC protein is associated with shorter overall survival independently of other gene alterations or BCL2 expression [42]. In addition to its many other roles, MYC controls expression of many metabolic genes, including those involved in 1C metabolism and all three enzymes of the serine synthesis pathway (SSP) [62]. In accordance with this, we recently observed high levels of SSP enzymes in BL, low expression in CLL, and heterogeneous expression of these enzymes in DLBCL, where high expression was associated with reduced overall survival [32]. The importance of the SSP for DLBCL has been highlighted by other work identifying copy number gains genes encoding SSP enzymes, along with genomic amplification of serine hydroxyl-methyltransferase (SHMT), an enzyme that converts serine to glycine. [63]. Interestingly, B-cell lymphoma cell lines are sensitive to blockade of glycine synthesis by SHMT inhibition due to defective import of glycine from the microenvironment [64]. In addition to its regulation of glycolysis, the SSP and 1C metabolism, MYC also controls glutaminolysis [65]. Notably, a recent study has also implicated the mitochondrial lysine deacetylase sirtuin-3 (SIRT3) in promoting glutamine metabolism to support DLBCL growth [66]. SIRT3 was found to be overexpressed in DLBCL where it was linked to inferior clinical outcomes, and DLBCL cells were observed to be dependent on SIRT3 for their proliferation, survival and self-renewal. Deletion or inhibition of SIRT3 impaired glutamine metabolism via the TCA cycle, which resulted in reduced acetyl-CoA levels, induction of autophagy and death of the cells, and resultant attenuation of lymphoma growth in vivo [66].

### 3.4. Chronic Lymphocytic Leukemia (CLL)

There have also been several recent advances in the understanding of the metabolism of CLL. Initial observations noted an increase in the mitochondrial mass of CLL cells when compared with healthy B-cells, with accompanying increases in mitochondrial respiration, mitochondrial reactive oxygen species and oxidative stress [67,68,69,70]. These findings, along with reports regarding increased expression of molecules involved in fatty acid metabolism, suggested that CLL cells rely predominantly on OXPHOS [71]. However, while this may well be the case for CLL cells circulating in the peripheral blood, it is becoming increasingly understood that CLL cells alter their metabolism when they enter the lymph nodes. It is well-documented that most CLL proliferation occurs within “proliferation centers” in CLL lymph nodes, where it is aided by contact with stromal cells, dendritic cells, T cells, cytokines and BCR-engagement within the CLL microenvironment [72]. Proliferation centers in CLL lymph node biopsies show increased expression of MYC and markers of major metabolic pathways such as glycolysis, SSP and OXPHOS [32,73,74]. A variety of mechanisms have been observed to drive these metabolic alterations including stromal cell contact, BCR signaling, NOTCH-MYC signaling and hypoxia [73,75,76,77,78]. Recently, a combined analysis of peripheral blood CLL cells from patients at baseline and after 3 months of treatment with ibrutinib, together with in vitro experiments to mimic lymph node microenvironment using BCR or CD40 stimulation, suggested that metabolic changes involving purines, glucose and glutamate, with emphasis on amino acids as a TCA fuel, could play role in CLL cells in lymph nodes [79]. Furthermore, the authors found that treatment with ibrutinib suppressed MYC target genes induced with BCR/CD40 stimulation. This is important clinically, as drugs that inhibit BCR-signaling also cause CLL cells to move out of the lymph nodes into the peripheral blood resulting in an increased lymphocytosis [80]. Therefore, it is likely that part of the clinical success of BCR signaling inhibitors is due the “metabolic reprogramming” that is induced by the expulsion of CLL cells from the pro-glycolytic microenvironment of the lymph nodes, along with direct inhibition of the metabolic changes induced by BCR signaling [73]. Removal of CLL cells from the nurturing lymph node microenvironment may have other important effects on CLL metabolism as stromal cells also provide cysteine to chronic lymphocytic leukemia cells to help support glutathione production and antioxidant defense, making the cancer cells more resistant to ROS-inducing therapy [81]. This may be modulated by the presence/absence of functional wild-type p53, due to its role as a negative regulator of glycolysis [82]. Notably we observed a decreased dependency on BCR signaling for metabolic activity in CLL harboring deletions of chromosome 17p (which includes the *TP53* gene), which is known marker of worse prognosis in CLL [73]. The Bcl-2 inhibitor venetoclax is a further example of how a clinically efficacious drug, originally developed to target a non-metabolic pathway, can have metabolic effects. Notably, resistance to this agent in B-cell lymphoma cell lines and primary CLL samples can be decreased by inhibiting OXPHOS or glutamine import, offering the potential for future combinatorial strategies [79,83].

## 4. Anti-Metabolic Therapies for Lymphoma

The emerging importance of metabolism in lymphoma has highlighted the potential for metabolic enzymes and pathways to be therapeutic targets (Figure 2). This is not a new concept as drugs such as methotrexate, which inhibits the folate cycle and 1C metabolism, has been used for decades as the backbone of protocols for the prophylaxis and treatment of central nervous system lymphoma [84]. The metabolic impact of drugs targeting the PI3K/Akt pathway has been a particular focus over the last 10 years, with several inhibitors of this pathway having been developed and tested [19]. Inhibitors of the p110δ isoform have had the greatest success clinically due to this isoform being expressed only in hematopoietic cells [85,86]. Idelalisib was the first agent to be approved for the treatment of CLL, small lymphocytic lymphoma (SLL) and FL in 2014. The overall response rates in a heavily pre-treated population of patients with these indolent lymphomas was 57%, with 6% of patients achieving complete remission with median progression free survival (PFS) of 11 months [85]. Widespread clinical usage has been hampered by the toxicity profile of these patients, with grade 3 adverse events including neutropenia, diarrhea, transaminitis and pneumonitis, but they remain an option for certain groups of patients [87]. Other PI3K inhibitors have subsequently been developed that target the PI3Kδ isoform along with isoforms and kinases including copanlisib (PI3Kδ and PI3Kα), duvelisib (PI3Kδ and PI3Kγ) and umbralisib (PI3Kδ and casein kinase-1 epsilon). While they all show efficacy and have been approved for low-grade lymphomas, their efficacy in DLBCL has generally been disappointing. They also show comparable toxicities to idelalisib, with copanlisib also having other serious side effects such as hyperglycemia and hypertension (reviewed in [88]).

The effector of PI3K signaling and cellular metabolic sensor, mTORC1, is dysregulated in several lymphomas (including DLBCL and FL) and has been studied extensively as a therapeutic target [19,89]. Its inhibitors, rapalogs, unfortunately showed limited clinical efficiency in DLBCL and have only been approved for refractory/relapsed mantle cell lymphomas so far [89]. As rapalogs have a cytostatic effect as monotherapy, it has been suggested that the cellular metabolic adaptations induced by the treatment might in turn be effectively targeted by co-inhibition of signaling or metabolic pathways upstream or downstream of mTORC1 [89]. These include already-approved drugs such as ibrutinib to inhibit BCR signaling or antimetabolites that inhibit nucleotide synthesis as well as drugs that inhibit mitochondrial respiration or glutaminolysis [89]. In one small but provocative study, three out of four patients with refractory DLBCL achieved complete remission (CR) in response to a combination of the mTOR inhibitor temsirolimus, with L-asparaginase and metformin to inhibit glutamine and the respiratory chain respectively [60].

Another major class of antimetabolic agent that has been undergoing clinical testing for lymphoma are inhibitors of the monocarboxylate transporter (MCT) family, typically MCT1. These are transmembrane proteins that mediate the bidirectional transport of lactate (as well as other substrates such as pyruvate, short-chain fatty acids and ketones) in and out of cells [90]. As MCTs allow the removal of excess lactate produced from increased glycolytic activity, inhibition results in accumulation of intracellular lactate, causing cellular acidification and impeding glycolytic flux and potentially NAD+ regeneration [91]. In addition, inhibition of lactate export into the cellular microenvironment could be expected to inhibit the growth of any other cells that are using this metabolite as fuel source [39]. A study of 120 DLBCL samples and 10 BL samples found that they all expressed MCT1 protein, but generally lacked expression of MCT4 [91]. A phase I expansion study of the MCT1 inhibitor AZD3965 in patients with relapsed/refractory DLBCL and BL has recently been reported [88,92]. Unfortunately, while the results of AZD3965 as a monotherapy were somewhat disappointing with only one out of eleven patients achieving a CR, this patient’s tumor did exhibit reduced FDG uptake as early as day 3 of therapy, consistent with the predicted impact of AZD3965 on glycolytic flux [92]. Interestingly, preclinical work with lymphoma cell lines demonstrated that resistant cells manifested more OXPHOS that could be overcome by combining the inhibition of MCT1 and with inhibitors of respiratory complex I, an approach that could be tested in the clinic [91,93].

Several recent reviews have summarized new developments in targeting cancer metabolism and the range of metabolic inhibitors undergoing preclinical and clinical testing [94,95]. Among these, many inhibit enzymes that have been identified as potential therapeutic targets in lymphomas such as hexokinase 2 (HK2), which catalyzes the first committed step in glycolysis, or mitochondrial dihydroorotate dehydrogenase, which is important for pyrimidine synthesis [96,97]. Even though there has been a lot of interest into targeting HK2, particularly as its expression has been shown to positively correlate with FDG-uptake in lymphoma, no inhibitor has yet entered clinical trials [98]. However, new strategies to inhibit HK2 activity or to displace the enzyme from its normal location on the mitochondrial outer membrane are being investigated [94,99]. A number of other drugs are undergoing clinical testing in patients with solid and hematological cancers including lymphomas, including the glutaminase inhibitor telengenastat (CB-839), the mitochondrial complex I inhibitor IM156, and devimistat (CPI-613), a lipoate analog that inhibits mitochondrial oxidative metabolism (NCT02071888, NCT03272256, NCT04217317).

Another area of potential future therapeutic innovation regards the impact of dietary modulation on lymphoma. Numerous studies spanning over a century of research have shown that caloric restriction can slow growth of a variety of tumors in rodent models, and in patients who have reduced intake due to anorexia or gastric bypass surgery [100]. More recently, the focus has turned to a number of other dietary interventions including the ketogenic diet, and diets that restrict specific amino acids such as serine, glycine or methionine [101]. The serine–glycine-free diet is of particular interest in light of observations showing that it could significantly prolong the survival of mice with *MYC*-driven lymphoma, and that these mice also respond to inhibition of the serine synthesis pathway [32,102]. Indeed, a further potential strategy would be to combine metabolic inhibitors and dietary intervention. The development of PI3K inhibitors was challenging until the emergence of idelalisib, a specific inhibitor of PI3Kδ. The suspected reason for lack of clinical efficacy of inhibitors of the alpha subunits (PI3Kα) is that this mediates insulin signaling, resulting in increased glucose–insulin feedback, which is sufficient to re-activate the PI3K signaling targeted in the first place [103]. With this in mind, dietary regimens such as the ketogenic diet could aid the therapeutic response to inhibitors targeting PI3Kα [103,104]. This exemplifies the huge variety of possible inhibitor–diet combinations that could be developed to treat relapsed/refractory lymphoma in the future [105]. The combination of particular diets with metabolic inhibitors could result in synergy enabling lower doses of drugs to be effective while mitigating against resistance [105]. Other examples include the combination of serine restriction with inhibitors of mitochondrial respiration, methionine restriction with chemotherapy and histidine supplementation to potentiate the effect of methotrexate [105,106,107]. The other part of this therapeutic strategy would then be to “personalize” it to a patient’s particular tumor, for example, to match therapy to individual genetic lesions or to the metabolic subtypes discussed above [57,94]. While several preclinical studies reported differential drug sensitivities between the glycolytic BCR-DLBCL and the OXPHOS subtypes, these concepts remain in their infancy and much research, both preclinical and clinical, is required [57,58,59,108].

## 5. Conclusions

Research advances over the last couple of decades have made a huge impact on therapy and improved patient outcome in some of the B-cell malignancies, particularly CLL, while comparable changes in treatment for high-grade lymphomas such as DLBCL and BL have lagged behind. There is no doubt that advances in immunotherapy, including chimeric antigen receptor-modified T-cells and bispecific antibodies are making an impact and will have an expanding role in the future. Inhibitors of kinases involved in BCR signaling such as ibrutinib and idelalisib, along with BCL2 inhibitors such as venetoclax have underpinned the paradigm shift in CLL treatment. However, resistance mutations have been identified that either affect the ability of the drug to bind to the target kinase or activate downstream signaling to bypass the pharmacological blockade. One of biggest advantages of anti-metabolic therapy is that it blocks a process reliant on a physical metabolite rather than a signal, making it harder or even impossible for cancer cells to bypass. However, several challenges remain. One of the major hurdles in translation of metabolic inhibitors into the clinic is the disparity between in vitro and in vivo observations. This can be documented by conflicting results on mTORC1 activation in B-cell lymphoma cell lines versus primary human lymphoma samples, along with differential effects observed with mTORC1 inhibitors in cell culture and in clinical trials [89]. This may reflect inadequacies in the model systems being used. The concentration of metabolic substrates in traditional cell culture media was designed to keep passaged cells alive, and do not necessarily represent the concentrations found in the body. Furthermore, it is even more challenging to model differences in the concentrations that a cell may encounter when trafficking between different body compartments and to maintain a particular nutrient at the same concentration, which is homostatically maintained in a living organism [109]. These factors can have major impacts on observations meaning that results derived from an in vitro model system or even a murine model are not necessarily translatable into humans [110]. Use of more accurate and physiological culture media and careful validation in vivo and in patients will help to address this [110,111]. However, the major strides that have been made in the understanding of normal and malignant B-cell metabolism over the last few years, mean that a clinical breakthrough of a novel metabolic therapy for lymphoma is now within reach.

## Figures and Tables

**Figure 1 cancers-14-05552-f001:**
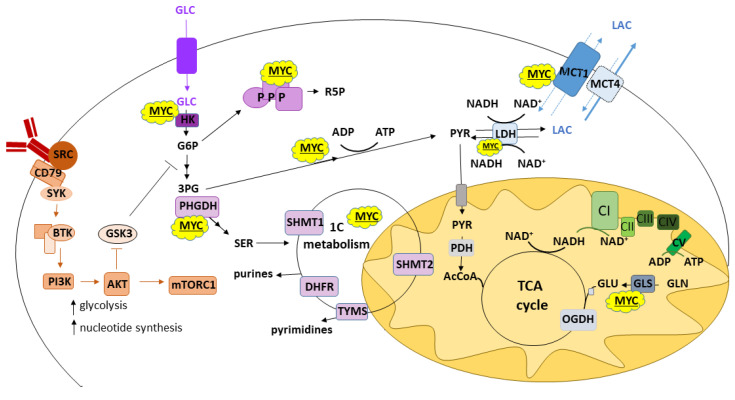
Metabolic pathways in B-cell lymphoma. Imported glucose (GLC) is metabolized via glycolysis with multiple branching pathways utilizing glycolytic intermediates and pyruvate (PYR): glucose-6-phosphate (G6P) can be metabolized via the pentose phosphate pathway (PPP) to ribose-5-phosphate (R5P); 3-phosphoglycerate (3PG) can be used for serine (SER) and glycine (not shown) synthesis and contribute to one-carbon (1C) metabolism and the synthesis of purines and pyrimidines; glycolytic pyruvate can be converted to lactate (LAC) (while regenerating NAD^+^) and exported out of the cell or metabolized further in mitochondria and enter the tricarboxylic acid (TCA) cycle as acetyl coenzyme A (AcCoA). Glutamine (GLUT) conversion to glutamate (GLU) is one of anaplerotic pathways that can feed the TCA cycle. These metabolic pathways are regulated at multiple levels and via several modes such as kinase signaling triggered by BCR activation (shown in orange) and transcription factor MYC (shown as yellow clouds). Other abbreviations used in the figure: Bruton tyrosine kinase (BTK), phosphoinositide-3-kinase (PI3K), glycogen synthase kinase 3 (GSK3), mammalian target of rapamycin complex 1 (mTORC1), hexokinase (HK), phosphoglycerate dehydrogenase (PHGDH), serine hydroxymethyl transferase 1/2 (SHMT1/2), dihydrofolate reductase (DHFR), thymidylate synthase (TYMS), monocarboxylate transporter 1/4 (MCT1/4), lactate dehydrogenase (LDH), pyruvate dehydrogenase (PDH), glutaminase 1 (GLS1), oxoglutarate dehydrogenase (OGDH), individual complexes comprising oxidative phosphorylation (CI-V; shown in green).

**Figure 2 cancers-14-05552-f002:**
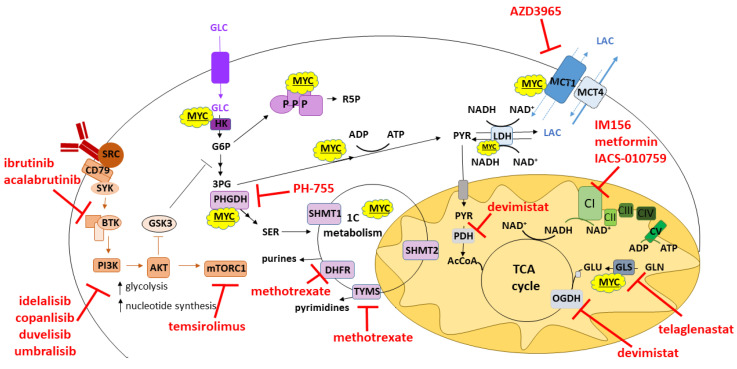
Metabolic inhibitors and their targets in B-cell lymphoma. Pro-proliferative kinase signaling downstream of the BCR can be targeted by inhibiting BTK (e.g., ibrutinib or acalabrutinib), PI3K (e.g., idelalisib, copanlisib, duvelisib, umbralisib) and mTORC1 (e.g., temsirolimus). Enzymes involved in glucose and 1C metabolism can be inhibited at the point of serine synthesis via PHGDH (e.g., PH-755) and tetrahydrofolate regeneration by dihydrofolate reductase (methotrexate) and thymidylate synthesis by thymidylate synthase (e.g., methotrexate). Lactate efflux can be blocked by inhibition of MCT1 by AZD3965. Mitochondrial carbon metabolism can be targeted via inhibition of glutaminase (e.g., telaglenastat) and by inhibiting lipoate dependent enzymes pyruvate dehydrogenase and oxoglutarate dehydrogenase (e.g., devimistat). Mitochondrial oxidative phosphorylation and NADH oxidation can be targeted by specific inhibition of respiratory complex I (e.g., IM156, metformin, IACS-010759).

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
