# Peer review of "Advances in Understanding of Metabolism of B-Cell Lymphoma: Implications for Therapy"

_cancers, 2022, doi:10.3390/cancers14225552_

Round 1

Reviewer 1 Report

Kluckova et al has written a review on metabolism in B-cell lymphomas. Overall, the review is consistent and of great interest for lymphoma interested researchers. I have some minor comments. 

1.     The simple summary is supposed to be a simple summary, not just a shorter form of the abstract. 

2.     Line 46: It is true that germinal center-derived DLBCL originate from germinal center B-cells, but all DLBCL are not germinal center-derived.  

3.     Why is venetoclax only mentioned in the conclusions and not before?

Author Response

Kluckova et al has written a review on metabolism in B-cell lymphomas. Overall, the review is consistent and of great interest for lymphoma interested researchers. I have some minor comments. 

We thank the Reviewer for their comments regarding the interest of the manuscript. We have addressed their comments as below (in italics).

  1. The simple summary is supposed to be a simple summary, not just a shorter form of the abstract. 

We have amended the simple summary to make the article more accessible for the reader and to distinguish it more clearly from the abstract.

  1. Line 46: It is true that germinal center-derived DLBCL originate from germinal center B-cells, but all DLBCL are not germinal center-derived.  

We have amended this sentence to reflect that not all types of DLBCL are germinal centre derived (line 47).

  1. Why is venetoclax only mentioned in the conclusions and not before?

We thank the Reviewer for this comment. We have included a couple of sentences regarding the metabolic impact of venetoclax on B-cell lymphoma cell lines and primary CLL samples (lines 307-311).

Reviewer 2 Report

Thank you very much for providing an opportunity to review the article titled “Advances in understanding of metabolism of B-cell lymphoma: implications for therapy” by Katarina Kluckova and co-authors.

In this review article, Katarina Kluckova and co-authors discuss the current understanding of the normal germinal centre metabolism and how this relates to the metabolism of germinal centre derived B cell lymphomas and also discuss the rational for employing therapeutic approaches to target the actionable nodes of these major metabolic pathways. 

Comments:

  1. In addition to MYC and mTOR, HIF-1alpha plays a major role in energy metabolism in normal B cells as well as in B cell lymphomas. It would be ideal to adequately discuss the important metabolic pathways controlled by HIF-1 in the normal as well as the discussed lymphomas. This is especially relevant since, both Myc and HIF often cross-talk in several of the energy metabolism pathways (ref: Gordan J.D. Bertout J.A. Hu C.J. Diehl J.A. Simon M.C.Cancer Cell. 2007; 11: 335-347). 
  1. Correct the name of GSK3 to Glycogen Synthase Kinase 3, Lane 94.

Author Response

Reviewer 2

Thank you very much for providing an opportunity to review the article titled “Advances in understanding of metabolism of B-cell lymphoma: implications for therapy” by Katarina Kluckova and co-authors.

In this review article, Katarina Kluckova and co-authors discuss the current understanding of the normal germinal centre metabolism and how this relates to the metabolism of germinal centre derived B cell lymphomas and also discuss the rational for employing therapeutic approaches to target the actionable nodes of these major metabolic pathways. 

We thank the Reviewer for their kind comments regarding the manuscript. We have addressed these directly as below.

Comments:

  1. In addition to MYC and mTOR, HIF-1alpha plays a major role in energy metabolism in normal B cells as well as in B cell lymphomas. It would be ideal to adequately discuss the important metabolic pathways controlled by HIF-1 in the normal as well as the discussed lymphomas. This is especially relevant since, both Myc and HIF often cross-talk in several of the energy metabolism pathways (ref: Gordan J.D. Bertout J.A. Hu C.J. Diehl J.A. Simon M.C.Cancer Cell. 2007; 11: 335-347). 

We thank the Reviewer for this comment. We have added some comments regarding the importance of HIF in normal GC B cells (lines 97-103), Burkitt lymphoma (lines 196-199) and DLBCL (lines 243-245), and have included the suggested reference.

  1. Correct the name of GSK3 to Glycogen Synthase Kinase 3, Lane 94.

 We have corrected GSK3 to Glycogen Synthase Kinase 3 as suggested.

Reviewer 3 Report

This provocative manuscript by Kluckove et al. summarizes the metabolic programming of B cells in the germinal center and during different types of lymphoma and highlights its implications in B cell lymphoma therapy. Overall, the manuscript is well-written. The authors have justified the important concept’s presentation and incorporated appropriate references.

Author Response

Reviewer 3

This provocative manuscript by Kluckove et al. summarizes the metabolic programming of B cells in the germinal center and during different types of lymphoma and highlights its implications in B cell lymphoma therapy. Overall, the manuscript is well-written. The authors have justified the important concept’s presentation and incorporated appropriate references.

We thank the Reviewer for their positive comments regarding the manuscript and for their review.